# Zebrafish Motile Cilia as a Model for Primary Ciliary Dyskinesia

**DOI:** 10.3390/ijms22168361

**Published:** 2021-08-04

**Authors:** Andreia L. Pinto, Margarida Rasteiro, Catarina Bota, Sara Pestana, Pedro Sampaio, Claire Hogg, Thomas Burgoyne, Susana S. Lopes

**Affiliations:** 1Paediatric Respiratory Medicine, Primary Ciliary Dyskinesia Centre, Royal Brompton & Harefield NHS Trust, London SW3 6NP, UK; A.Pinto@rbht.nhs.uk (A.L.P.); c.hogg@rbht.nhs.uk (C.H.); t.burgoyne@ucl.ac.uk (T.B.); 2CEDOC, NOVA Medical School, Rua Câmara Pestana nº 6, 6-A, Edifício CEDOC II, 1150-082 Lisboa, Portugal; margarida.rasteiro@nms.unl.pt (M.R.); Catarina.bota@nms.unl.pt (C.B.); Sara.pestana@nms.unl.pt (S.P.); Pedro.sampaio@nms.unl.pt (P.S.); 3Department of Life Sciences, NOVA School of Science and Technology, 2825-149 Caparica, Portugal; 4Department of Paediatrics, Imperial College London, London SW3 6LY, UK; 5Institute of Ophthalmology, University College London, London EC1V 9EL, UK

**Keywords:** transmission electron microscopy, motile cilia, left–right organizer, olfactory pit, electron tomography, animal model, zebrafish

## Abstract

Zebrafish is a vertebrate teleost widely used in many areas of research. As embryos, they develop quickly and provide unique opportunities for research studies owing to their transparency for at least 48 h post fertilization. Zebrafish have many ciliated organs that include primary cilia as well as motile cilia. Using zebrafish as an animal model helps to better understand human diseases such as Primary Ciliary Dyskinesia (PCD), an autosomal recessive disorder that affects cilia motility, currently associated with more than 50 genes. The aim of this study was to validate zebrafish motile cilia, both in mono and multiciliated cells, as organelles for PCD research. For this purpose, we obtained systematic high-resolution data in both the olfactory pit (OP) and the left–right organizer (LRO), a superficial organ and a deep organ embedded in the tail of the embryo, respectively. For the analysis of their axonemal ciliary structure, we used conventional transmission electron microscopy (TEM) and electron tomography (ET). We characterised the wild-type OP cilia and showed, for the first time in zebrafish, the presence of motile cilia (9 + 2) in the periphery of the pit and the presence of immotile cilia (still 9 + 2), with absent outer dynein arms, in the centre of the pit. In addition, we reported that a central pair of microtubules in the LRO motile cilia is common in zebrafish, contrary to mouse embryos, but it is not observed in all LRO cilia from the same embryo. We further showed that the outer dynein arms of the microtubular doublet of both the OP and LRO cilia are structurally similar in dimensions to the human respiratory cilia at the resolution of TEM and ET. We conclude that zebrafish is a good model organism for PCD research but investigators need to be aware of the specific physical differences to correctly interpret their results.

## 1. Introduction

Motile cilia are centriole-derived organelles, surrounded by a membrane and containing microtubules formed by protofilaments that can be longer or shorter, depending on the number of tubulin molecules they contain, according to the function they perform [1,2,3]. This structure is called the ciliary axoneme or ciliary shaft [4], a transversal term throughout species, due to the well-conserved ciliary structure [3]. The axonemal ultrastructure in eukaryotic cells segregates into two significant patterns: 9 + 2, in which nine microtubule doublets organise around two microtubules in the centre known as the central pair (CP) complex; the 9 + 0 organisation, in which the CP is absent [5,6]. Moreover, there are two major types of motile ciliated cells: cells that produce from dozens to several hundred 9 + 2 motile cilia and cells that only generate one cilium.

Motile monocilia-lacking CPs were described in the node of mice [7,8,9]. These are very different from non-motile primary cilia (also with a 9 + 0 configuration) without dynein motor arms that can be found in almost every cell type having solo sensory functions and lacking the ability to generate movement [10,11]. On the other hand, Yu et al. in 2011 identified cilia with a 9 + 2 organisation but immotile, called kinocilia, in the hair cells of the inner ear that express *Foxj1b* [12] and in the ciliated receptor cells (primary sensory cells) of the zebrafish OP [13]. We can, thus, speculate that all combinations are present in nature, likely by adaptive evolution.

Regarding motile cilia dynamics, multiciliated cells that typically have cilia with a 9 + 2 configuration beat metachronically with a planar stroke to clear fluid or promote locomotion [14,15]. On the other hand, motile cilia lacking CP (9 + 0) have a specific movement pattern, described as planar rotation, circling or twisting. These 9 + 0 monocilia are present in the LRO in vertebrates such as mice or zebrafish [16,17,18], but reports on LRO cilia with 9 + 2 cilia and 9 + 4 also exist [7,18,19]. In cases of disease, multiciliated cells with cilia wholly or partially lacking the CP are frequently accounted as one of the phenotypes of Primary Ciliary Dyskinesia, an autosomal recessive disease of the motile cilia [20,21]. The rationale being that in the respiratory cilia, a rotational movement is not efficient in mucociliary clearance and is, therefore, causative of PCD symptoms.

Zebrafish became popular as an animal model in the 1980s, triggered by George Streisinger’s studies showing zebrafish as a genetically tractable organism, allowing a phenotypic characterisation of a large number of mutations that cause defects in a variety of organ systems [22]. As zebrafish are vertebrates, the translational interpretation of ciliary defects is very powerful, when compared, for instance, to *Chlamydomonas reinhardtii*, where the defects from cilia impairment are usually spotted as a lack of locomotion [23], although not exclusively. Zebrafish helped shed light on the role of specific genes in human diseases, as their genome has been widely studied, providing insight into their human orthologues [24]. Amongst the many advantages of zebrafish, we will stress the fact that zebrafish embryos contain cilia in nearly every cell type, and their organogenesis defects can be easily characterised using brightfield and fluorescent microscopy as the zebrafish embryo and larvae are mostly transparent until all major organ systems are formed. Many of the organs and tissues of zebrafish are similar to those of humans and 70% of genes are shared [24]. Motile cilia have been well described in the Kupffer’s vesicle, the zebrafish LRO, between 3 and 14 somite stages (ss) of development and in the OP at 48–72 hpf, amongst other locations [25].

Zebrafish, as almost all teleosts, have three types of olfactory receptor neurons (ORNs): ciliated ORNs, microvillous ORNs and crypt cells with both cilia and microvilli [26,27]. The bottom of the OP is coated with ORNs, each one with protruding non-motile primary cilia that contain olfactory receptors [28]. On the other hand, the rim is surrounded by a layer of cuboidal multiciliated cells protruding bundles of motile cilia [28]. These microvillous ORNs, and ciliated ORNs have similar morphological and molecular contents to the microvillous and ciliated ORNs of higher vertebrates [29]. Moreover, genetic engineering methodologies, including CRISPR/Cas9, have been successfully used in editing the zebrafish genome and have greatly facilitated the generation of zebrafish mutant models mimicking human ciliopathies [30,31,32].

PCD is a genetic disorder distinguished by recurrent infection in the lower and upper respiratory tract [33], reduced fertility and laterality problems (50% of PCD patients show *situs inversus*) [34]. Ultrastructural defects in motile cilia or a reduced cilia number are known to cause PCD [34,35]. Currently there are more than 50 genes identified that can cause PCD [36]. Almost all PCD genes show homologous genes in zebrafish, as shown in Table 1. Despite previous research on zebrafish cilia [13,18,26,37], no study fully characterised the cilia ultrastructure of the OP and LRO. With this work, we aim to determine the similarities and the significant differences between zebrafish and human motile cilia, comparing OP multiciliated cells and embryonic LRO monociliated cells. Using transmission electron microscopy (TEM), we confirmed the heterogenic configuration of the zebrafish OP cells. We also showed by electron tomography (ET) the variable presence of a CP in the LRO monocilia, a feature hypothesised in a previous publication by Tavares et al. [18].

## 2. Results

### 2.1. Zebrafish Motile Cilia

Zebrafish have motile cilia in many of its organs, that are present since the early stages of development, as depicted in the diagram from Figure 1.

Olstad et al. have previously shown the flow generated by the OP motile cilia [39]. Whereas Sampaio et al. (2014) and Tavares et al. (2017) have extensively described the rotational and wavy fashion beat of monocilia of the LRO [16,18]. These motile monocilia are essential to generate flow for the determination of left–right asymmetry [16,18,40]. We confirmed the localization and distribution of cilia in the wild-type (WT) LRO and OP of zebrafish by immunofluorescence (IF). To better understand how to orient the embryo for the subsequent TEM embedding and sectioning, we generated 3D blend projections and surfaces as shown in Figure 2 for the OP. We consider that this study greatly helped the TEM work. Confocal microscopy is useful but limited to 180 nm lateral and 500 nm axial resolution and is not appropriate for ultrastructural studies. Therefore, we next used TEM to access the ultrastructure of the respective organ cilia.

Next, we evaluated the cilia beat frequency for both organs, so that, in the future, researchers can compare it with zebrafish disease models for PCD. CBF in the LRO ranges from 15 to 50 Hz was evaluated by CiliarMove [41]. This software creates a heatmap for CBF that allowed us to unequivocally detect in a very visual way different cilia in the same focal plane beating differently as depicted by the blue, green and yellow ciliary colour codes (Figure 3A’). This raised an intriguing question as to what is the function of this spatial heterogeneity in the LRO CBF? Compared to the OP, where all cilia beat around 20 Hz, in the green colour code (Figure 3B’), it is tempting to speculate that monocilia in the LRO do not coordinate their CBF for a reason that may relate to the establishment of asymmetry. In addition to the intra-LRO cilia CBF variability, we also detected an inter-embryonic variability greater in the LRO than in the OP (Figure 3C).

### 2.2. Different Cilia in the Zebrafish Olfactory Pit

To characterise the ultrastructural pattern of cilia in the OP, a detailed investigation was conducted by means of examining cilia from five dpf zebrafish by TEM. The quantitative analysis of three different WT animals (N = 113 cilia; values show mean ± standard deviation) showed that 60% (± 1) of cilia had a 9 + 2 arrangement with dynein arms present, and 23% (± 2) of cilia presented absent or incomplete dynein arms (most notably in the outer dynein arm (ODA)) (Figure 4 and Table 2). We further showed the heterogenicity of cilia within specific regions of the OP. When determining the localisation of the different ciliary ultrastructural types, cilia observed in the peripheral OP had dynein arms reflecting their motile function, whereas cilia in the central OP had absence of ODA and inner dynein arms (IDA). A TEM analysis allowed the quantification of the heterogenicity of cilia in a specific region of the OP. Cilia observed in the OP had a 9 + 2 motile morphology in a more peripheral area and a 9 + 2 with an ODA and IDA absence towards the centre of the OP.

The percentages of missing ODA and IDA assessed by quantitative methods [42] (as shown in Table 2) were compared using Student’s *t*-test in both zebrafish and human samples for a significance analysis. The comparisons between WT zebrafish OP cilia and healthy control patients concerning the presence of ODAs and IDAs were significantly different (ODA *** *p* ≤ 0.001; IDA * *p* ≤ 0.05, Table 2 and Figure 5). In healthy humans, ODAs were rarely missing, contrary to the observed findings in the central region of the zebrafish OP. Therefore, for research purposes of modelling PCD using zebrafish OP cilia, one should consider cilia from the OP periphery.

Next, cross-sections from the LRO cilia were also analysed by TEM (Figure 5). A limited number of cilia were observed due to the monociliated nature of the LRO cells. The ultrastructure of these cilia, as shown previously [18], had dynein arms in WT embryos (n > 100 cilia) and some cilia show an absent or partial configuration of the CP as shown in Figure 5.

To compare the structure of the microtubule doublets of zebrafish LRO cilia against human respiratory cilia, we performed ET to generate 3D reconstructions. After assessing the cilia from the LRO by ET, some concern regarding the size of the ODA was raised, as some variation in the size of the outer dynein arms was visible, suggesting it might be smaller. To clarify, we used the software Chimera [43] (USFC, California) to measure the volume of the ODA which was normalised to the total volume of the microtubule doublet (MTD). We analysed the volume of the MTD from the LRO of four different WT zebrafish and compared them to the MTD of three human control respiratory cilia (Figure 6). No significant differences were found between the two samples, indicating that LRO cilia and human airway cilia have similar ODA volumes (student’s test, *p* > 0.05).

### 2.3. The Application of Transmission Electron Tomography for the Study of the Central Pair

Despite the visible heterogenicity of the CP in the LRO cilia, there was a need to provide more evidence to validate these findings. The LRO is an organ well embedded in the embryo and hard to access. It has a spherical-like shape and, depending on the plane of sectioning, cilia will be in different orientations. Additionally, LRO cells are monociliated [44]. To acquire well-oriented cross-sections of many cilia in the LRO is hard and produces sparse results, even in the event of finding an axoneme, the probability of it being well-orientated is low. We were able to collect tomograms from six cilia from two different WT zebrafish LROs (Figure 7 embryo 1: cilium 1/4 to 4/4; embryo 2: cilium 1/2 and 2/2). We investigated the prevalence of the CP using a z-projection of the slices of the tomogram, which combines all the slices to generate a single image. From this assessment, we could identify a heterogeneous pattern regarding the CP at different random locations in the LRO. Two cilia showed evidence of a CP, two cilia showed only one of the CP tubules (incomplete phenotype), and two cilia presented no visible CP (Figure 7).

## 3. Discussion

Cilia are complex organelles, and their dysfunction leads to many syndromes and diseases making them essential targets in biology and translational research [4,45]. PCD patients have a 50% chance of developing incorrect left–right placement of internal organs, but the difficulty of studying human embryonic tissues provides the need to use model organisms that resemble human organogenesis and have conserved evolutionary patterns. The *Chlamydomonas reinhardtii* and *Leishmania donovani* promastigotes are good models [30,46,47], easy to maintain and proliferate in captivity. However, most of these models lack internal organs and organ-to-organ communication, crucial for the study of PCD. Therefore, vertebrates such as mouse and zebrafish are better models to study PCD since their genomes are well studied and contain protein-coding genes with human orthologues [25]. This feature has been used for PCD research in many studies [48,49,50,51,52,53,54]. Some advantages of the zebrafish model are the transparency of the embryos for left–right related research allowing phenotypes to be followed in the same individuals from embryo to larva, low costs in maintenance and a high number of eggs progenies per cross. PCD has already been associated with more than 50 genes known to cause disease [4] and all of those genes are present in the genome of zebrafish [38] (Table 1). Despite being a good model, we show in this study that zebrafish have differences regarding cilia ultrastructure that need to be known to researchers in order to be used wisely (Table 3).

Ultrastructural methodologies have improved in the last ten years allowing for higher resolution images and better sample preparation techniques, leading to detailed structural studies of organelles [55]. The zebrafish Kupffer’s vesicle, the fish LRO, equivalent to the mouse node, is an important structure to perform left–right cilia research. However, this vesicle is embedded at the end of the notochord and it is challenging to access [8,16]. Even though the zebrafish is transparent, and the LRO is accessible by light microscopy, a more detailed analysis using conventional TEM can be problematic. The organ itself is not bigger than 70 µm in diameter (measured by TEM), but its presence is transient during the zebrafish development [16,17,56]. In addition, the LRO has an ovoid shape, leading to cilia with many different orientations when imaged by TEM from ultrathin cross-sections. Furthermore, the LRO is only composed of around 50 monociliated cells (assessed by light microscopy) [16,17]. Altogether, these features make the LRO a challenging organ to study by conventional TEM. On the other hand, the OP is a surface organ more easily accessed [37].

Our findings, that the OP has a heterogenic pattern of ciliary ultrastructure, namely, the presence of immotile cilia in the central region of the OP, coincide with what was already described by Reiten et al. [37]. At the time, the authors described a decrease in the beat frequencies in the centre of the OP, hypothesising that motile cilia in the nose of aquatic vertebrates were spatially organised to attract particles, showing no beating in the centre of the pit [37]. However, the authors did not undertake TEM analysis for the OP cilia [37]. The presented work sheds light on the lack of flow detected in the centre of the pit, given the immotile profile of the cilia nested in that specific area (lacking dynein arm motor proteins). Our findings and the work performed by Reiten seem to agree with observations previously published by Hansen (1998, 2005), in which a subgroup of ciliated sensory neurons are believed to cluster in the centre of the OP and to have a 9 + 2 configuration with no dynein motors. However, these previous studies were conducted in adult zebrafish, and extensive an TEM analysis was not undertaken to characterize the ultrastructure of these cilia [13,26]. Here, we, thus, revealed the presence of a patch of immotile cilia in the central OP region in five dpf zebrafish larvae.

Using ET, we were able to show that the MTD ODA from the LRO resembles the structure of the MTD ODA of the cilia found in the human airway. We also showed the presence of IDA and ODA in all cilia analysed, suggesting that all cilia may have the ability to generate movement. We have shown before by live imaging that motile and immotile cilia are present in the zebrafish LRO in a mixed pattern [18] contrary to the mouse LRO that shows immotile cilia at the periphery and motile cilia in the centre. In addition, we showed there is a heterogenic CP structure throughout the LRO cilia. Some cilia show a CP while others do not, although we cannot rule out that CP presence may be interrupted along each cilium. This feature shows that cilia in the zebrafish LRO are different from the ones in the human node. Namely, human LRO cilia are thought to be devoid of CP and radial spokes because mutations in these proteins do not lead to left–right defects in humans [57]. On the other hand, mutations in radial spoke proteins, such as Rsph9 in zebrafish, resulted in laterality defects [58]. Our present work explains why this occurs, and alerts researchers to this difference between humans and zebrafish LROs. It is essential that differences between humans and animal models are known for a correct interpretation of the translational research (summarized in Table 3).

Nevertheless, the differential presence of a CP in the zebrafish LRO suggests different beating patterns in the LRO. Different frequencies [16,17] for the LRO cilia were described before by our group [16] and confirmed here applying the CiliarMove software. CP variability may have a huge impact on cilia movement and, consequently, on fluid flow generation; thus, this study identified the need to establish the localization of each cilium with or without CP in the LRO. Only then, we can correlate it with regional differences in flow dynamics, which are important for proper left–right asymmetry. Another alternative hypothesis is that the CP might be partially present along the axoneme. A serial analysis of the LRO and its cilia is recommended in the future. Immunofluorescence with a CP marker and large volume EM will allow a better understanding of the distribution of different cilia and allow to map the heterogenic ciliary ultrastructure within the LRO.

In conclusion, zebrafish peripheral OP motile cilia resemble cilia found in the human airway and are accessible for manipulation and analysis. Zebrafish LRO cilia are a suitable model to test the left–right phenotypic effects of PCD mutations, but it is important to know that zebrafish LRO cilia have a CP.

## 4. Materials and Methods

### 4.1. Ethical Approval

The ethics review board from the Institute of Child Health/Great Ormond Street Hospital study, London (UK) approved these studies (08/H0713/82 15/10/2008). All subjects gave informed signed consent for genetic testing and surplus diagnostic samples to develop new diagnostic tools.

### 4.2. Animals

The zebrafish used in these experiments (AB WT line) were bred and kept in the CEDOC Nova Medical school fish facility under a controlled environment approved by the Direção Geral de Alimentação e Veterinária (DGAV). Embryos and larvae stages 8 to 10 ss and 2 to 5 days post fertilization (dpf), respectively, were removed from their chorion if needed and euthanised by being placed in a bleach solution or fixative for TEM and ET analysis.

### 4.3. Immunofluorescence and Image Analysis

Embryos were fixed in 4% PFA (paraformaldehyde) in PBS (phosphate buffer solution) overnight. The embryos were then removed from the PFA and washed five times with PBS, dechorionated, permeabilized with Proteinase K and fixed again in 4% PFA-PBS for 20 min. The samples were submitted to antigen retrieval using acetone for 7 min at −20 °C, and blocking was conducted using blocking solution (BS) (1% bovine foetal serum (BFS) in PBS) for 1 h. The primary antibody (mouse anti-acetylated alpha-tubulin) was diluted at 1:400 in BS and incubated with the embryos overnight at 4 °C with agitation. The embryos were incubated with secondary antibody overnight (anti-mouse Alexa 546, 1:500) in a solution with equal amounts of PBS and DAPI (4′,6-diamidino-2-phenylindole). Samples were fixed in 4% PFA in PBS and were mounted onto glass slides. Images were taken using a two-photon microscope (40× water objective, NA 0.80, 2.5 optical zoom GaAsP detector). The acquired 3D datasets were processed and volume rendered in blend view using Imaris (Bitplane) v.9.5.0. Imaris was also used to construct the 3D surface of the OP by selecting the desired ROI in the OP marked by DAPI and cilia are shown in a 3D MIP.

### 4.4. Transmission Electron Microscopy

Embedding for TEM was performed in zebrafish embryos (9–10 ss) and larvae (5-dpf) from WT AB. The animals were whole-mount fixed with 2.5% glutaraldehyde (Sigma) 0.1 M sodium cacodylate (Sigma) for at least 48 h and washed in the same buffer. Afterward, they were post-fixed with 1% aqueous osmium tetroxide (EMS) and stained en bloc with 1% aqueous uranyl acetate (Agar Scientific). Samples were dehydrated in increasing ethanol concentrations, infiltrated with propylene oxide and embedded in EPON 812 (EMS). Polymerisation of the EPON was performed at 60 °C for 24 h in bottleneck BEEM capsules (Agar Scientific). Under a stereoscope, the blocks containing the embryos and larvae were analysed, and the best ones (based on their viability and orientation) were selected for re-embedding in the desired orientation. The tip of the blocks containing the samples was chopped and the blocks containing the fragments were re-embed in coffin-type silicon moulds. For the larvae, only the head was dissected and re-embedded in coffin-type silicon moulds in an anterior–posterior position, with the OP facing the plane of sectioning. The 9–10 ss embryos were re-embedded lateral to obtain longitudinal sections along the wider axis to fully access the LRO, the somites and the yolk sac. The resin was polymerised in an oven at 60 °C for at least 24 h. Ultrathin sections (≥100 nm to be able to obtain at least one complete repeat of the IDA) were obtained in a Reichert Ultracut E ultramicrotome (Leica^®^) collected to 1% formvar-coated copper slot grids (Agar scientific), stained with 2% aqueous uranyl acetate (UA) and lead citrate (LC) (Reynold recipe) and examined in a JEOL 1400 plus transmission electron microscope at an accelerating voltage of 120 kV. Digital images were obtained using an AMT XR16 bottom mid-mount digital camera (AMT). The sections were systematically analysed using AMT600 software, and several high- and low-magnification images of cilia throughout the whole LRO and the OP were acquired. Considering the high abundance of cilia found in the OP, these were counted using a custom-made software developed to quantify cilia for PCD diagnosis [42]. Patient samples were embedded for TEM and analysed as previously described in Andreia L. Pinto et al. [42], following a standard clinical diagnostic sample preparation and assessment procedure.

### 4.5. Electron Tomography

Sections from the same batch of animals were acquired for ET analysis; thick sections of 100–120 nm were collected into 1% formvar-coated copper slot grids (Agar scientific), stained with 2% aqueous UA, LC (Reynold recipe) and 10 nm colloidal gold (EMS), and examined in a JEOL 1400 plus transmission electron microscope at an accelerating voltage of 120 kV. For dual-axis ET, selected areas at a magnification of 25.000× were tilted from −60 to 60 degrees in the two axes (the grid was rotated 90 degrees between the collecting tilt series) using a high-tilt specimen holder. SerialEM was used for the automatic acquisition of the images. Cilia cross-sections of the LRO and the OP were assembled in a tomogram using IMOD 3Dmod [59], and both axes were combined for the generation of an electron tomogram of the region of interest. Sub-tomographic averagings were performed using IMOD PEET in both organs of WT fish. For the 3D analysis of the structures, UCSF Chimera was used. Chimera is a program for the interactive visualisation and analysis of molecular structures and related data. We opened the data in Chimera with the averaged ciliary features and produced a model showing the molecular anatomy of the structure, and we used Chimera to measure the volumes of the ODA and MTD [60].

### 4.6. Data Analysis

In order to better study the discrepancies between the experimental groups, Student’s *t*-test was performed, and the significance was considered for values below 0.05.

## Figures and Tables

**Figure 1 ijms-22-08361-f001:**
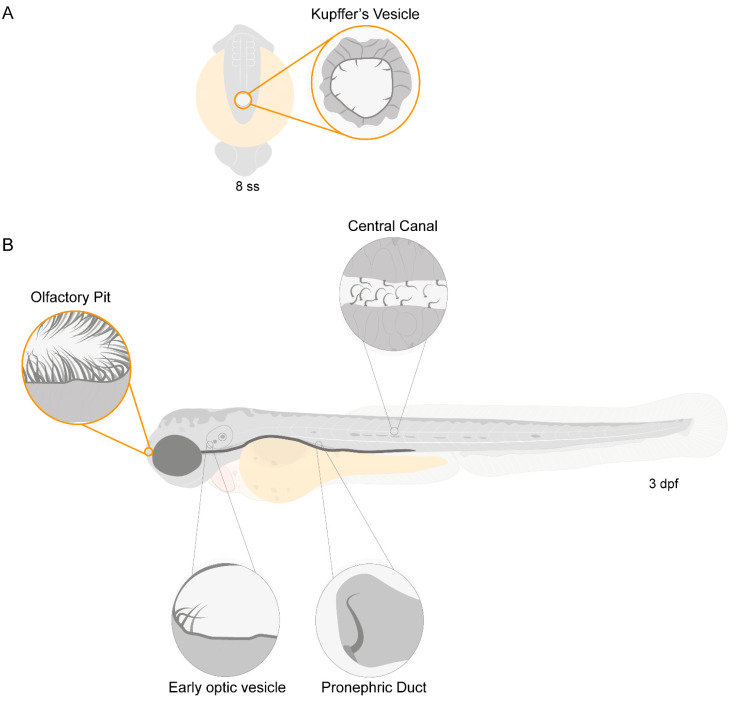
Motile ciliated structures in zebrafish. Schematic representation of (**A**), a zebrafish embryo at 8 somite stages (ss) highlighting Kupffer’s vesicle (KV); (**B**) a 3-day post-fertilization (dpf) zebrafish larva indicating structures with motile cilia. Olfactory pit (OP) and KV cilia analysed in the present study are circled in orange.

**Figure 2 ijms-22-08361-f002:**
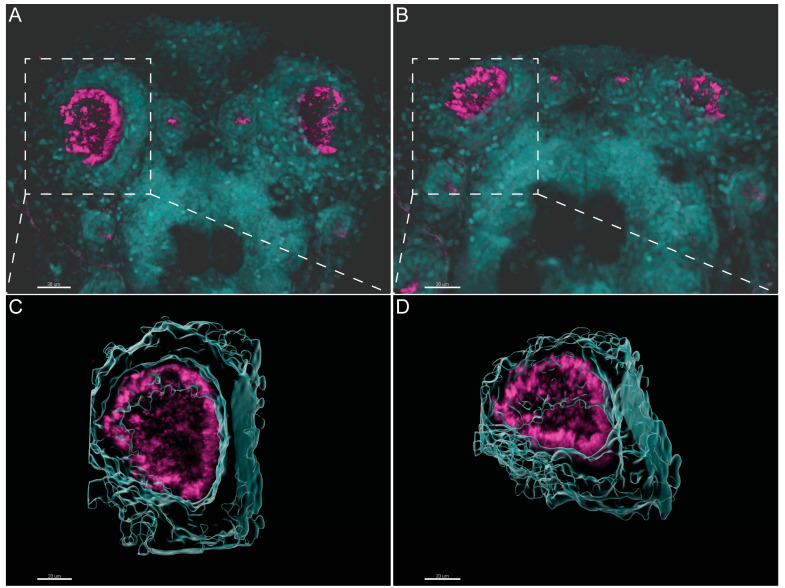
Three-dimensional imaging analysis helps to orient and localize the OP of zebrafish for TEM studies. (**A**,**B**). Immunofluorescent labelling with anti-acetylated α-tubulin shows the distribution of multiciliated cells in the OP of 4 dpf larvae. Software Imaris (Bitplane) v.9.5.0 allowed 3D blend reconstructions of 2 different OPs from 2 different larvae. (**C**,**D**) 3D surface reconstructions from the respective OPs revealing the concave morphology of the organ when rotated. Anti-Acetylated α-tubulin immunofluorescence in magenta and DAPI in cyan. Scale bar 20 μm.

**Figure 3 ijms-22-08361-f003:**
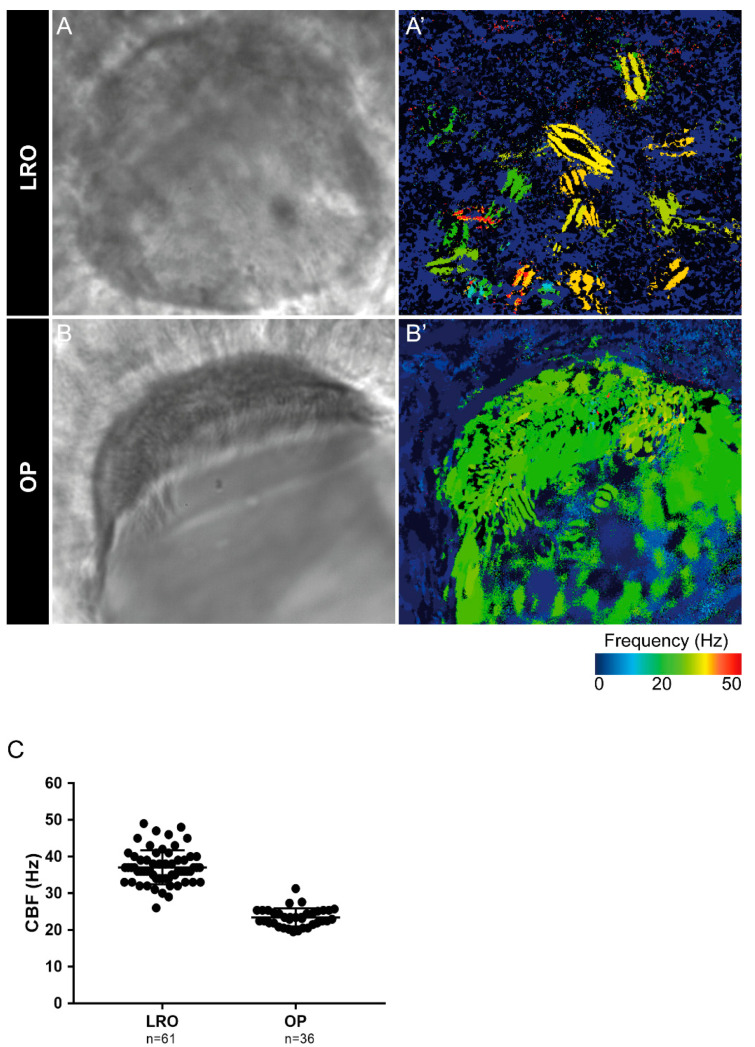
Cilia beat frequency (CBF) evaluation for zebrafish cilia. (**A**,**A’**) Full LRO heatmap showing several monocilia at one plane beating at different frequencies (20–50 Hz). (**B**,**B’**) OP heatmap showing multiciliated cells with cilia beating at more homogeneous frequencies (around 20 Hz). CBF was measured using the software CiliarMove [41] (**C**) Quantification of CBF from n = 61 LROs and n = 36 OPs from embryos at 10 ss and 4 dpf larvae, respectively.

**Figure 4 ijms-22-08361-f004:**
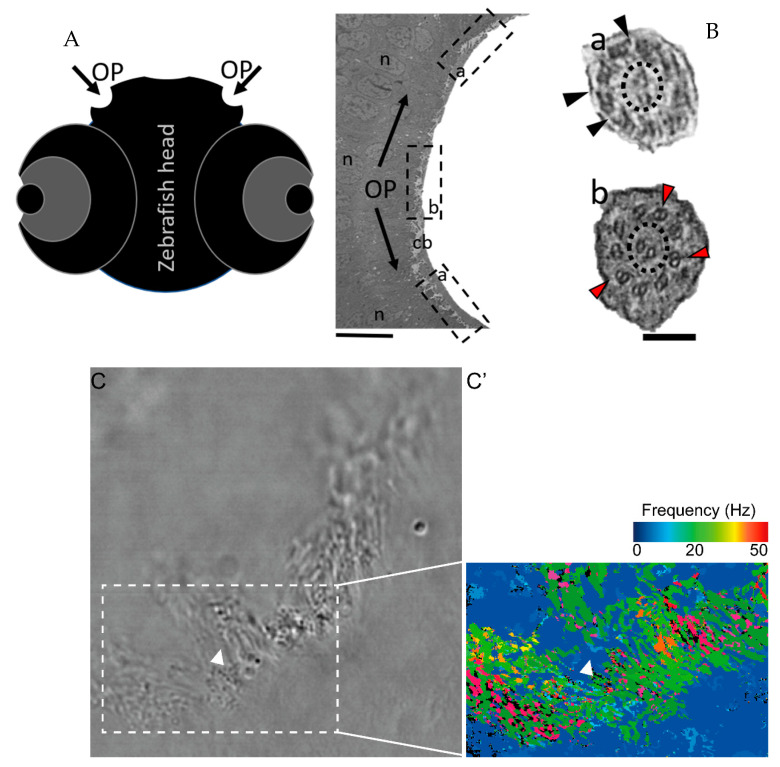
The zebrafish OP has two different types of cilia. (**A**) Schematic representation of 5 dpf zebrafish head, structures of interest are marked with arrows—olfactory pits (OP). (**B**) TEM low magnification image showing a cross-section across an OP of a WT zebrafish, a bowl-shaped structure containing multiciliated cells and cilia in several orientations. In the periphery of this pit (dotted boxes a and inset), cilia containing classical motile structure 9 + 2 with dynein arms (black arrowheads). In the most internal region of the OP (dotted box b and inset) we detected cilia with 9 + 2 ultrastructural arrangement without dynein arms (red arrowheads); this pattern was visible in WT embryos (n = 3). (**C**) Snapshot from a movie of beating OP cilia and respective heatmap by software CiliarMove (**C’**), highlighting a region of immotile cilia (arrowhead) that was coincident with the region detected by TEM in B. n—nucleus; cb—cilia border; OP—olfactory pit. Thin bar 6 µm, thick bar 100 nm.

**Figure 5 ijms-22-08361-f005:**
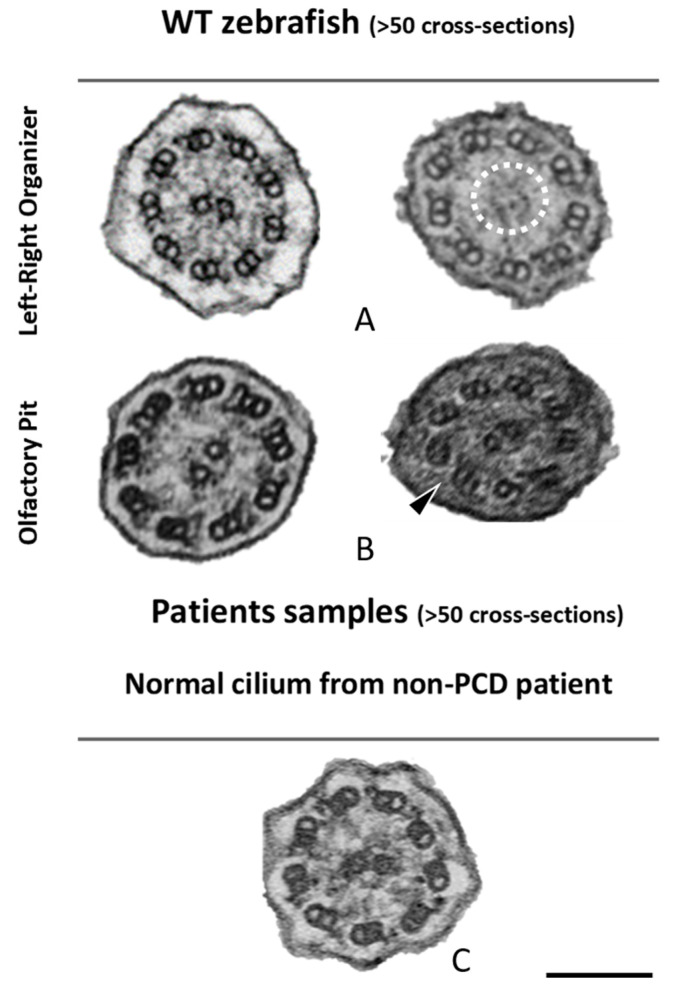
Ciliary ultrastructural comparison between the LRO, OP and human respiratory cilia. Variations of the axonemal arrangement of cilia in (**A**) the LRO of 10 somite stages WT zebrafish embryo and (**B**) the OP of 5 dpf WT zebrafish larvae. (**C**) Example of a human respiratory cilium cross-section from the airway. Arrowheads indicate missing ODA and the dotted circle shows missing CP. Scale bar 100 nm.

**Figure 6 ijms-22-08361-f006:**
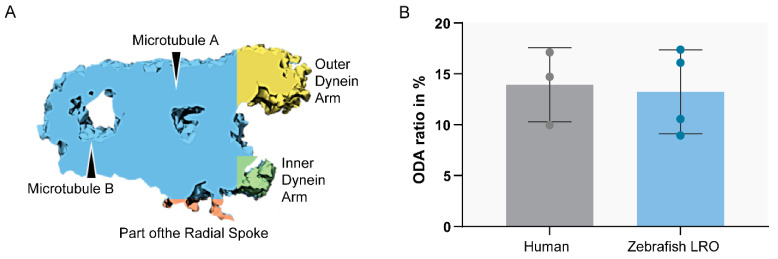
Ratio between ODA and MTD volume. Sub-tomographic averaging of the MTD of WT zebrafish cilia, showing ODA and IDA, tubules (**A**,**B**) and a portion of the radial spoke. The two sample groups were not significantly different (*p* > 0.05, Student’s *t*-test). N = 3 human patients and n = 4 zebrafish LROs.

**Figure 7 ijms-22-08361-f007:**
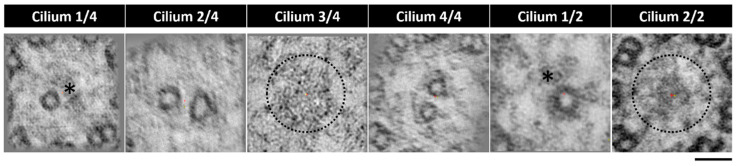
Tomogram z-projection confirms central pair heterogeneity. CPs from two different LROs (n = 2) and from six different cilia, showing variation in the presence of the CP. Cilium 1/4 and 1/2 showing only one tubule present, cilium 2/4 and 4/4 showing both tubules present and cilium 3/4 and 2/2 showing no tubules in the central pair region. * missing tubule, incomplete CP; dotted circle missing CP. Bar 50 nm.

**Table 1 ijms-22-08361-t001:** Human-Zebrafish homologue PCD genes as described in Ensemble.org [38].

PCD Gene	Zebrafish Transcript Name	Zebrafish Transcript ID
*DNAH5*	*dnah5-201*	ENSDART00000123150.4
	*dnah5-202*	ENSDART00000191818.1
*CCDC114 (ODAD1)*	*ccdc114-201*	ENSDART00000023745.8
*ARMC4 (ODAD2)*	*cr847789.1-201*	ENSDART00000186851.1
	*armc4-201*	ENSDART00000077453.5
	*armc4-204*	ENSDART00000170018.2
	*armc4-203*	ENSDART00000153115.2
	*armc4-202*	ENSDART00000152887.2
*TTC25 (ODAD4)*	*ttc25-201*	ENSDART00000080946.5
*DNAH9*	*dnah9-201*	ENSDART00000160926.2
*DNAH11*	*dnah11-201*	ENSDART00000148294.4
	*dnah11-202*	ENSDART00000020821.10
	*dnah11-203*	ENSDART00000138744.4
*DNAI1*	*dnai1.2-201*	ENSDART00000080431.5
	*dnai1.2-202*	ENSDART00000142468.3
	*dnai1.1-205*	ENSDART00000170205.2
	*dnai1.1-201*	ENSDART00000160163.2
	*dnai1.1-204*	ENSDART00000169676.2
	*dnai1.1-202*	ENSDART00000163063.2
	*dnai1.1-203*	ENSDART00000165798.2
*DNAI2*	*dnai2a-201*	ENSDART00000162579.2
	*dnai2a-202*	ENSDART00000164199.2
	*dnai2b-203*	ENSDART00000188726.1
	*dnai2b-201*	ENSDART00000003339.9
	*dnai2b-202*	ENSDART00000188648.1
*DNAL1*	*dnal1-203*	ENSDART00000188500.1
	*dnal1-202*	ENSDART00000156182.2
	*dnal1-201*	ENSDART00000043651.7
*TXNDC3 (NME8)*	*nme8-201*	ENSDART00000163684.2
*CCDC103*	*ccdc103-201*	ENSDART00000075493.4
	*ccdc103-202*	ENSDART00000132293.2
*CFAP298 (C21orf59)*	*cfap298-201*	ENSDART00000051197.6
	*cfap298-202*	ENSDART00000130093.3
	*cfap298-203*	ENSDART00000181950.1
*CFAP300 (c11orf70)*	*cfap300-201*	ENSDART00000151109.2
	*cfap300-202*	ENSDART00000192737.1
*DNAAF1 (LRRC50)*	*dnaaf1-201*	ENSDART00000145762.4
	*dnaaf1-203*	ENSDART00000173909.2
	*dnaaf1-202*	ENSDART00000173853.2
*DNAAF2 (KTU)*	*dnaaf2-201*	ENSDART00000167840.2
*DNAAF3*	*dnaaf3l-201*	ENSDART00000079233.5
*DNAAF4 (DYX1C1)*	*dnaaf4-201*	ENSDART00000165855.2
*DNAAF5 (HEATR2)*	*lo018183.1-201*	ENSDART00000194031.1
*DNAAF6 (PIH1D3)*	*pih1d3-201*	ENSDART00000056375.5
	*pih1d3-203*	ENSDART00000145388.3
	*pih1d3-202*	ENSDART00000136858.2
	*pih1d3-204*	ENSDART00000183524.1
	*pih1d3-205*	ENSDART00000191761.1
*LRRC6*	*lrrc6-203*	ENSDART00000188883.1
	*lrrc6-202*	ENSDART00000132346.3
	*lrrc6-201*	ENSDART00000075347.5
*RPGR*	*rpgrb-201*	ENSDART00000088624.5
	*rpgrb-202*	ENSDART00000124471.3
	*rpgrip-201*	ENSDART00000138541.3
	*rpgrip-203*	ENSDART00000190953.1
	*rpgrip-202*	ENSDART00000179003.2
	*rpgrip1l-202*	ENSDART00000185324.1
	*rpgrip1l-201*	ENSDART00000126326.5
*SPAG1*	*spag1b-201*	ENSDART00000101207.5
	*spag1a-202*	ENSDART00000185960.1
	*spag1a-201*	ENSDART00000130537.3
*ZMYND10*	*zmynd10-201*	ENSDART00000017413.10
	*zmynd10-202*	ENSDART00000189261.1
	*zmynd10-203*	ENSDART00000183251.1
*CCDC39*	*ccdc39-202*	ENSDART00000190769.1
	*ccdc39-201*	ENSDART00000169709.2
*CCDC40*	*ccdc40-202*	ENSDART00000169752.2
	*Ccdc40-201*	ENSDART00000164275.2
	*Ccdc40-203*	ENSDART00000182267.1
*TTC12*	*ttc12-201*	ENSDART00000156234.2
	*ttc12-202*	ENSDART00000157380.2
*CCDC65 (DRC2)*	*ccdc65-201*	ENSDART00000043946.8
	*ccdc65-202*	ENSDART00000177219.2
*CCDC164 (DRC1)*	*drc1-201*	ENSDART00000061829.5
*GAS8*	*gas8-202*	ENSDART00000170982.2
	*gas8-201*	ENSDART00000165126.2
*CFAP221*	not found in ZF	
*DNAJB13*	*dnajb13-204*	ENSDART00000148093.3
	*dnajb13-201*	ENSDART00000063365.6
	*dnajb13-203*	ENSDART00000139097.2
	*dnajb13-202*	ENSDART00000133505.2
*HYDIN*	*hydin-201*	ENSDART00000143265.4
	*hydin-202*	ENSDART00000145701.2
	*hydin-203*	ENSDART00000169861.2
	*bx571975.1-201*	ENSDART00000185269.1
*NME5*	*nme5-201*	ENSDART00000060998.6
*RSPH1*	*rsph1-201*	ENSDART00000160273.3
	*ct573248.3-201*	ENSDART00000181186.1
*RSPH3*	*rsph3-202*	ENSDART00000128823.5
	*rsph3-201*	ENSDART00000103394.3
*RSPH4a*	*rsph4a-201*	ENSDART00000097340.5
*RSPH9*	*rsph9-201*	ENSDART00000010903.8
*STK36*	*stk36-201*	ENSDART00000086765.5
	*stk36-202*	ENSDART00000139065.2
*SPEF2*	*spef2-201*	ENSDART00000159718.2
	*spef2-202*	ENSDART00000168984.2
*CFAP57*	*cfap57-201*	ENSDART00000080900.6
	*cfap57-202*	ENSDART00000149309.3
*LRRC56*	*lrrc56-202*	ENSDART00000161369.2
	*lrrc56-201*	ENSDART00000150364.2
*GAS2L2*	*gas2l2-201*	ENSDART00000112744.4
*NEK10*	*nek10-201*	ENSDART00000155162.2
*OFD1*	*ofd1-201*	ENSDART00000000552.12
*CCNO*	*fq311924.1-201*	ENSDART00000158096.2
*FOXJ1*	*foxj1b-201*	ENSDART00000126676.2
	*foxj1b-203*	ENSDART00000181942.1
	*foxj1b-202*	ENSDART00000153327.2
	*foxj1a-201*	ENSDART00000157772.2
	*foxj1a-202*	ENSDART00000168280.2
*MCIDAS*	*cu633857.1-201*	ENSDART00000192716.1

**Table 2 ijms-22-08361-t002:** Full TEM assessment and quantification of defects in cilia from the OP in wildtype zebrafish and human healthy controls. N = 3 zebrafish OPs and n = 113 cilia were examined, values shown are mean ± standard deviation.

	Disarranged Cilia (%)	Dynein Arms Assessment (%)			
Counted > 50 Cilia		Both Arms Present	ODA Missing	IDA Missing	Both Arms Missing
WT zebrafish (n = 3)	14 (±8)	62 (±1)	23 (±2)	14 (±2)	14 (±2)
Human control (n = 3)	3 (±1)	93 (±12)	1 (±1)	3 (±6)	2 (±4)

**Table 3 ijms-22-08361-t003:** Comparison between zebrafish and human cilia from the LRO and from the OP/nasal cilia.

	LROs	OP and Nasal Cilia
Zebrafish	Monociliated cells Most cilia present central-pair and radial spokes Intermixed distribution of immotile cilia (20%) and motile cilia (80%) from 7 ss onwards CBF ranges from 15 to 40 Hz in the same embryo	Multiciliated cells Central region of the OP has immotile cilia All cilia have a CP and radial spokes CBF around 20 Hz Coordinated cilia beat pattern between cilia Produce water flow
Human	Monociliated cells Cilia are thought to be devoid of central-pair and radial spokes	Multiciliated cells All cilia are motile All cilia have a CP and radial spokes CBF ranging from (6.18–9.17 Hz at 25 °C [41] Coordinated cilia beat pattern between cilia Produce mucus flow

## Data Availability

The data presented in this study are available on request from the corresponding author.

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
