# Peer review of "Zebrafish Motile Cilia as a Model for Primary Ciliary Dyskinesia"

_ijms, 2021, doi:10.3390/ijms22168361_

Round 1
Reviewer 1 Report
This is an interesting manuscript about the utility of Zebrafish as a model for Primary Ciliary Dyskinesia (PCD). The data provided make a case for the use of Zebrafish in the study of PCD. The suggestions for changes are primarily for clarification.
Line 20: Insert (LRO) after left right organizer. This is the first place that this abbreviation should appear.
Line 23: Insert (OP) after Olfactory pit
Line 30: Insert the word “physical” before “differences”
Line 36: Are you indicating that the cilia are dynamic in their length or that cilia can be of different lengths depending on their purpose/location?
Line 69: Lack of cilia impairment has effects on more than motility in C. reinhardtii.
Line 77: Make more clear that LRO and Kupffer’s vesicle are the same thing in Zebrafish. I suggest that you not label the Figure 1 with “Kupffer’s Vesicle” and then have to re-explain that it is the same as LRO in the legend.
Line 81: insert OP and delete pit
Line 81: ORNs Plural
Table 2 and Figure 5: The N numbers are given here, but should also be in other figures and in the Methods.
Line 124, 131 – 132: These last lines seem out of place.
Line 146: ODA define for first time
Line 149: Delete “a”
Line 146, 150 and throughout: remove – from “first-time” Should be first time.
Line 177: Insert “low number of” before “multicliated”
Line 216: change to heterogeneous
Line 231: Put all genus and species names in italics
Line 234: throughout delete “so” and replace with something else like “therefore”
Line 240: change “to cause” and delete “to be causing”
Line 258: change to heterogeneous
Line 271: delete “thus”
Line 282: Explain why you mention gene rsph9
Line 297: change to peripheral
Line 353: Patient samples. Where are PCD patient samples used?
Discussion: Right from the abstract and going forward, the authors warn that it is important to be aware of the differences between Zebrafish and PCD or other model organisms. It is incumbent upon the authors to provide a more detailed description of these differences. A more systematic description of PCD at the ciliary ultrastructure and function scale is needed and a more systematic comparison with the findings of this study. This study could be highly quoted in future research but too much is left to the reader to organize and make the take away conclusions.

Author Response
Dear Reviewer,
Thank you for your comments. We have modified the manuscript throughout according to your suggestions. We tried to clarify all the statements that you pointed out. Corrections are indicated in the document as track changes and with comments. As you suggested that we should make it more obvious for the reader what are all the differences and similarities between zebrafish and humans we included an extra Table3 in the discussion to summarize these differences in a more easy and visual way.
Thank you very much for your comments,
Best regards,
Susana
Reviewer 2 Report
Pinto et al. promotes the use of zebrafish as a model for PCD. They provide an Ensemble 2020 based table that shows the zebrafish orthologues for the ~50 genes known to cause PCD in humans. While they argue for the usefulness of zebrafish for studies of PCD, one of the main points of this work is to highlight some of the ciliary differences that one must be aware of in order to properly analyze and interpret your data. To that end they provide an analysis of cilia ultrastructure in two commonly used ciliated tissues, the LRO and the olfactory pit (OP). In the OP they provide useful data that shows that ciliary beating is low or negligent at the center of the pit. Importantly, they find that cilia in this area sometimes lack either or both outer and inner dynein arms. They further identify a significant amount of heterogeneity in cilia from the LRO ranging from cilia with 0, 1 or the normal two central pair microtubules. Based on the heterogeneity in cilia structure they urge the field to be careful. Overall I think the paper is well written but relatively light on new and useful data.
Comments:
In general, much of the paper reads like a review in both a good way because they did a nice summary of the literature, but also in a bad way:
Figure 1 is a cartoon (fine but no data)
Figure 2 is simply showing cilia in tissues that are well known to have cilia (which doesn’t add much)
Figure 3 is showing ciliary beat frequency that is also well characterized and doesn’t add much.
The heart of the paper is figures 5 and 7. In Figure 5 they provide nice quantification of the type of heterogeneity they see and where they see it. However, the LRO is much more challenging, which they appropriately describe. However, while I appreciate the challenges of performing EM tomography it is also hard to interpret results with such a low n (total of 6 cilia) and no geographical information, which is really important as the field has proposed in other systems that there are motile cilia in the center of nodes and immotile ones at the periphery. It would be nice if this work could integrate better with that model.
Minor:
“However, they lack internal organs”. C Elegans does not lack internal organs.
Not that I want to bash any model, but when discussing the strength of zebrafish as a model for PCD it might be nice to note that hydrocephalus is very common in mouse, but not humans, so zebrafish might be more similar to humans than mouse in some regards.
Author Response
Dear Reviewer,
Thank you very much for your comments. We have corrected the text according to your recommendations and agree with you that some figures do not bring much novelty as Figure 2. To correct this problem we have changed Figure 2 to include new data that was useful for the TEM study. We have replaced Figure 2 by a 3D reconstruction of the OP that is new. We consider this should be included because it shows how 3D reconstructions aid to the TEM block orientation in subsequent TEM samples. We deleted the KV images because similar work was already published by us and others in other studies (as you mentioned). Regarding Figure 1 we would like to keep it because its aim is to contextualize the broader PCD researcher. It focuses the reader in the motile cilia organs and shows where they are in the zebrafish embryo and larva.
Figure 3 has new data as it applies CiliarMove to the zebrafish for the first time and compares the two organs in many embryos and larvae (new data). It brought to our attention that at the same KV focal plane we can find different CBFs (new data) in opposition to the OP organ. The rest of the Figures also contain new data.
Regarding the LRO geographical information, the cilia were analysed at random places, according to their good/bad orientation for TEM, only well oriented cilia could be studied. Nevertheless, you are correct, we could have given the position of these few cilia in the LRO context. I am afraid this was not done and it is something we intend to do in the future by using immunostaining for the CP and using confocal microscopy instead (now that we know the CP may be present or absent). However, in zebrafish it is described by us and other labs that immotile cilia are intermingled with motile cilia, at a proportion of 20% of immotile to 80% of motile at 7-8 somites (I have explained this in the text now). This is a very interesting fact. They are indistinguishable and express foxj1a as well as present dynein arms. In Tavares et al 2017 (eLife) we dedicated a study to identifying what could be rendering these cilia immotile or motile. Her12 seems to be involved and this is still ongoing research in the lab. We have corrected all the mistakes pointed out and appreciate your remarks. We tried to make clear what are the new insights this paper brings and added a Table3 with the differences between the zebrafish model and human LRO and nasal cilia.
Best regards,
Susana